# Using Er:YAG laser to remove lithium disilicate crowns from zirconia implant abutments: An *in vitro* study

Janina Golob Deeb[1], Sompop Bencharit[2,3,4]*, Nishchal Dalal[5], Aous Abdulmajeed[2], Kinga Grzech-Leśniak[6]

**1** Department of Periodontics, School of Dentistry, Virginia Commonwealth University, Richmond, Virginia, United States of America, **2** Department of General Practice, School of Dentistry, Virginia Commonwealth University, Richmond, Virginia, United States of America, **3** Department of Oral and Maxillofacial Surgery, School of Dentistry, Virginia Commonwealth University, Richmond, Virginia, United States of America, **4** Department of Biomedical Engineering, College of Engineering, Virginia Commonwealth University, Richmond, Virginia, United States of America, **5** School of Dentistry, Virginia Commonwealth University, Richmond, Virginia, United States of America, **6** Department of Oral Surgery, Wroclaw Medical University, Wroclaw, Poland

☯ These authors contributed equally to this work.

\* sbencharit@vcu.edu

**Data Availability Statement:** All relevant data are within the paper and its Supporting Information files.

## Abstract

### Background

When implants are restored with cement-retained restorations, prosthetic retrievability can be difficult and often requires sectioning using rotary instruments. Sometimes repeated removals of a cement-retained implant crown are needed such as for treatment of peri-implantitis or immediate implant provisionalization. The purpose of this study was to evaluate the effect of erbium-doped yttrium aluminum garnet (Er:YAG) laser as a non-invasive treatment modality to remove lithium disilicate crowns from zirconia implant abutments following long-term cementation, repetitive debonding and re-cementation, and short-term retrieval.

### Material and methods

Twenty identical lithium disilicate crowns were cemented onto zirconia prefabricated abutments using composite resin cement. Ten cemented crowns were removed at 48 hours after cementation as a short-term group (ST), while another 10 were removed 6 months after cementation as a long-term group (LT). To mimicking repetitive recementation and retrieval, the LT crowns were then recemented and removed after 48 hours as a long-term recemention (LTR) group. The LTR crowns were then again recemented and removed after 48 hours as a long-term repeated recemention (LTRR) group. Er:YAG laser was used to facilitate the retrieval of these crowns. recorded and analyzed using ANOVA and t-test. The surfaces of the crown and the abutment were further examined using light microscopy and scanning electron microscopy (SEM). Temperature changes of the abutment and crown upto 10 minutes were also measured and statistically analyzed (paired t-test).

**Funding:** The authors received no specific funding for this work.

**Competing interests:** The authors have declared that no competing interests exist.

## Results

The average times of crown removal from zirconia abutments were 4 minutes (min) and 42 second (sec) in LT to 3 min 24 sec in LTR, and 3 min 12 sec in LTRR and ST groups. LTR took the longest time to remove, statistically (ANOVA and t-test, p < .001). No statistical differences were observed among the removal times of LTR, LTRR, and ST groups (t-test, p = .246, .246 and 1). SEM examination of the material surface showed no visual surface damaging from treatment with Er:YAG laser. The temperatures during irradiation ranged from 18.4˚C to 20˚C and 22.2˚C to 24.5˚C (Paired t-test, p < .0001) for the abutment and the crown during irradiation from 1 min to 10 mins.

## Conclusions

Long-term cementation can increase time in lithium disilicate crown removal from zirconia abutment using Er:YAG. Er:YAG laser is a non-invasive tool to remove cement-retained implant prostheses and should be considered as a viable alternative to rotary instruments.

## Introduction

Increasing patient demands and expectations in esthetics have driven the contemporary dental practice into all-ceramic restorations.[1–3] Dental implants present often the most desirable replacement of natural teeth in the esthetic zone.[4] Survival and longevity of dental implants along with the chance of multiple replacement of implant prostheses as well as future risks of peri-implantitis and fear of excess cement onto the implant cervical area have driven trend in screw-retained implant prostheses. However, it is not always possible to use a screw-retained prosthesis, especially in the esthetic zone where remaining residual ridge and occlusion as well as esthetics prefer the use of cement-retained implant prostheses.[5] Furthermore, superior esthetic outcomes for anterior implant restorations can be achieved with zirconia abutments together with all-ceramic crowns such as lithium disilicate.[6,7] Conventionally, titanium abutments are often used as an intermediate between the dental implants. Like other metals, titanium abutments can exhibit greyness or dark shade through a translucent lithium disilicate crown. Zirconia abutments are therefore often prescribed to offer white esthetics. Zirconia abutments have therefore an advantage over the metal abutment especially in patients with thin gingival biotype.[2,3,8,9]

Dental implants require long-term maintenance and after many years of use, replacement of prosthetic components is often indicated. Thus, retrievability of implant crowns and abutments is an important part of long-term dental implant therapy. Removal of cement-retained restoration can be a challenging task.[10–12] It often requires sectioning of the prosthesis using rotary instruments. Clinicians can spend a considerable amount of time and effort removing cement-retained prostheses. In many cases, it is desirable to be able to reuse the abutment and crown. These cases include various stages of peri-implantitis treatment, loose abutment screws, damaged restorations or provisional restorations that may require multiple removals of the crown/abutment to access the implant fixture.[13,14]

High strength modern all-ceramic materials such as lithium disilicate and zirconia can present a challenge in removal of the abutment/restoration without damaging the materials in order to be able to reuse the abutment/restoration. High-power lasers have been used for ablation, vaporization, disinfection and beneficial biological effects. [15,16] Pulsed erbium lasers,

such as erbium-doped yttrium aluminum garnet laser (Er:YAG) with 2940 nm wavelength and Erbium, chromium-doped yttrium, scandium, gallium and garnet (Er,Cr:YSGG) with 2780 nm wavelength, are theoretically useful for removing fixed prostheses. The peak absorption range of the water curve lies in the mid-infrared spectrum. Therefore, these lasers are effective for the treatment of hard tissue, in particular for the treatment of dental caries and bone defects.[17,18] Erbium laser can also be used in bone decortication and implant site preparation.[16,19] Recent studies demonstrated that Er:YAG can be used for metal and ceramic debonding of orthodontic brackets and veneer removal.[17,20] Removing a crown from a natural tooth abutment using Er:YAG laser has been shown to be effective and safe. Er:YAG can remove crown in minutes without damaging the restoration or causing excess thermal disturbance to the dental pulp.[19,21] A recent study suggested that removing a lithium disilicate crown from a titanium implant abutment can be done with minimal effort in a short period of time. Er:YAG applications showed no damage to the crown or abutment with little temperature changes.[22] However, there was no known effects of Er:YAG laser application for lithium disilicate crown removal from a zirconia abutment.

The goals of this study were to examine the feasibility of a non-invasive retrieval of a lithium disilicate crown from prefabricated zirconia implant abutments when the crowns were cemented for 6 months, for repeated cemented crowns and for newly cemented crowns. This simulated the clinical situation when a long-term cementation may be used such as in the case of immediate provisionalization, repeated crown removal in case of peri-implantitis treatment, or case with errors in new cementation and needed to be retrieved. It was also the aim to provide preliminary information on the parameter laser setting for future clinical applications of Er:YAG laser for lithium disilicate crown removal from an implant abutment. It was hypothesized that there was no statistically significant difference between removal times when removing a lithium disilicate crowns from zirconia abutments using Er:YAG laser in long-term and short-term cementation as well as from the repeated cementation.

## Materials and methods

The study protocol was adopted from the previous work.[22] A study cast from a patient with a single implant (4.5 mm platform Tapered Screw-Vent Implant, Zimmer Biomet) replacing mandibular right first premolar was used. A zirconia prefabricated implant abutments (Contour Abutment, Zimmer Biomet) were placed onto the study cast. An intraoral scanner (Emerald, Planmeca) was used to scan the abutment. Twenty identical lithium disilicate crowns (IPS e.max, Ivoclar Vivadent) were designed using Planmeca Romexis software (Planmeca) and milled out from an unsintered lithium disilicate block, iPS eMax (Ivoclar Vivadent). The crown was then sintered and glazed per manufacturer's recommendation (Ivoclar Vivadent). Then, the intaglio surface of the crown was pretreated for bonding (Monobond Etch & Prime, Ivoclar Vivadent). The abutments were hand-tightened onto an implant (4.7x16 mm, Tapered Screw-Vent Implant, Zimmer Biomet) and access opening in the abutment was filled with light body polyvinyl siloxane. The crown was then cemented onto the abutment using composite resin cement (Variolink Esthetics, Ivoclar Vivadent). The excess cement was removed and the cement was further light polymerized per manufacturer's recommendation (Ivoclar Vivadent). The crown/abutment was then kept in 100% humidity for at least 48 hours before retrieval attempt.

The cemented abutment-crowns were then divided into 2 groups, short-term (ST) and long-term (LT) cementation groups. The specimens in the ST group were kept for 48 hours before crown removal. The specimens in the LT group were kept for 6 months before crown removal. After the LT crowns were removed, the cement was removed with ultrasonic scalers

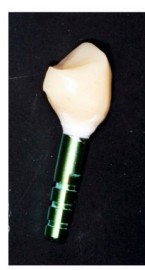

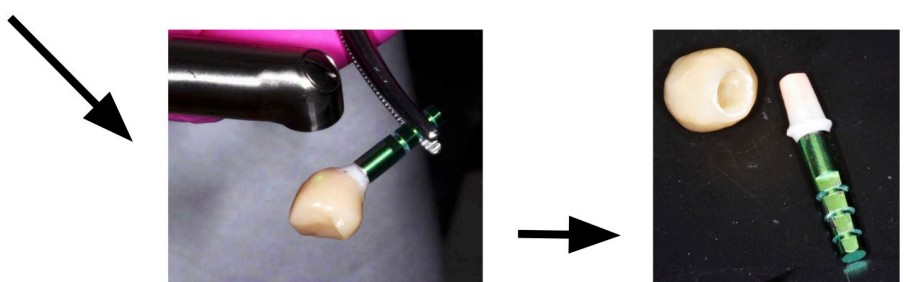

ST (48 hours)
LT (6 months)
LTR
(48 hours after debonding and recementation of LT)
LTRR
(48 hours after debonding and recementation of LTR)

Irradiation and debond attempts
incrementally every 30 sec.

**Fig 1. Study workflow.**

and steam cleaner. All LT crowns were then again undergoing cementation, kept for 48 hrs at 100% humidity, and removal was attempted at another time. This repeated removal crown was labelled as long-term recementation group (LTR). The removed LTR crowns were then undergoing cement removal, recementation, kept for 48 hrs at 100% humidity, and then again attempted removal. This last repeated removal crown was labeled as long-term repeated recementation group LTRR (Fig 1).

The samples were irradiated by means of laser Er:YAG at 2940nm (LightWalker, Fotona, Slovenia), using tipless handpiece (HO2, Fotona) at parameters: 300mJ; 15Hz; 4.5W, operation mode: QSP mode, with the distance 10mm, spot 0.9mm; and air/water spray at 2/2 with non-contact mode (Fig 1). Irradiation with the Er:YAG was directed perpendicular to the surface of the crown using a non-contact method with the distance between 5 to 10 mm from the restorative sample surface. The air/water spray was used throughout the irradiation process. The irradiation was performed axially on buccal, lingual and proximal surfaces, ~30–60 sec per surface. No irradiation was performed on the occlusal surface. Continuous motion of the laser handpiece on each axial surface was done to ensure even distribution of laser beam without stagnation. After all four surfaces were irradiated, the crown dislodgement was assessed using gentle finger pulling action. If the crown could not be dislodged after the first 3 min, another 30 sec of irradiation around the cervical area on all four surfaces would be employed. The crown dislodgement was then assessed once every 30 sec of extra irradiation. The total irradiation time of crown removal was recorded after the crown was debonded. The removal times for each group were analyzed for statistical significance using analysis of variance (ANOVA) and t-test (p<0.05).

After debonding, the underlying intaglio surface of the crown and the cameo surface of the abutment were inspected for visible damage using 2.5 power magnification (Sandy Grendel Loupes, Swiss Loupes). Visual inspection was done to examine if the cement was mainly

present on the crown intaglio surface or on the abutment. The surfaces were examined using light microscopy (Olympus SZX7, Tokyo, Japan) and then scanning electron microscopy (SEM) analyses (JEOL 6610LV, JEOL, Japan) to visualize surface damage. Secondary electron imaging (SEI) to topography of the structure analyze was applied. Additionally to enhance structural assessment and visualization of possible cracks, back-scattered electrons (BSE) was used, taking advantage of electron beam emerging from elastic scattering of deeper locations within the specimen. BSE was chosen for its sensitivity to material density and its ability to assess microstructural cracks.

To further examined the thermal changes of the crown-abutment from different irradiation times, the same cemented abutment-crown system was used similar to the previous work.[22] The crown was cemented onto abutment (n = 10) with an implant fixture (Fig 2) using a composite resin cement in the similar fashion as described earlier. A microthemal couple element (Adv. Thermocouple Therm. w/ RS232 Output Datalogger Type K- 800008, Super Scientific Works Pvt. Ltd.) was then placed onto the abutment and the crown. Er:YAG radiation with the same setting used to remove the crown used to radiate the abutment-crown was used to radiate the crown for 1, 2, 3, 4, 5, 6, 7, 8, 9, and 10 min, respectively. This time points were attempted to simulate minimal and maximal possible times used clinically. Temperature at each time point were recorded.

## Results

The average time of crown removal from zirconia abutments range from 4 min 42 sec in LT to 3 min 24 sec in LTR, and 3 min 12 sec in LTRR and ST (Table 1 and S1 File). There was a statistically significant difference among groups (ANOVA, p < .001). LT group had statistically the longest time to remove (t-test, p < .001). No statistical differences were observed among

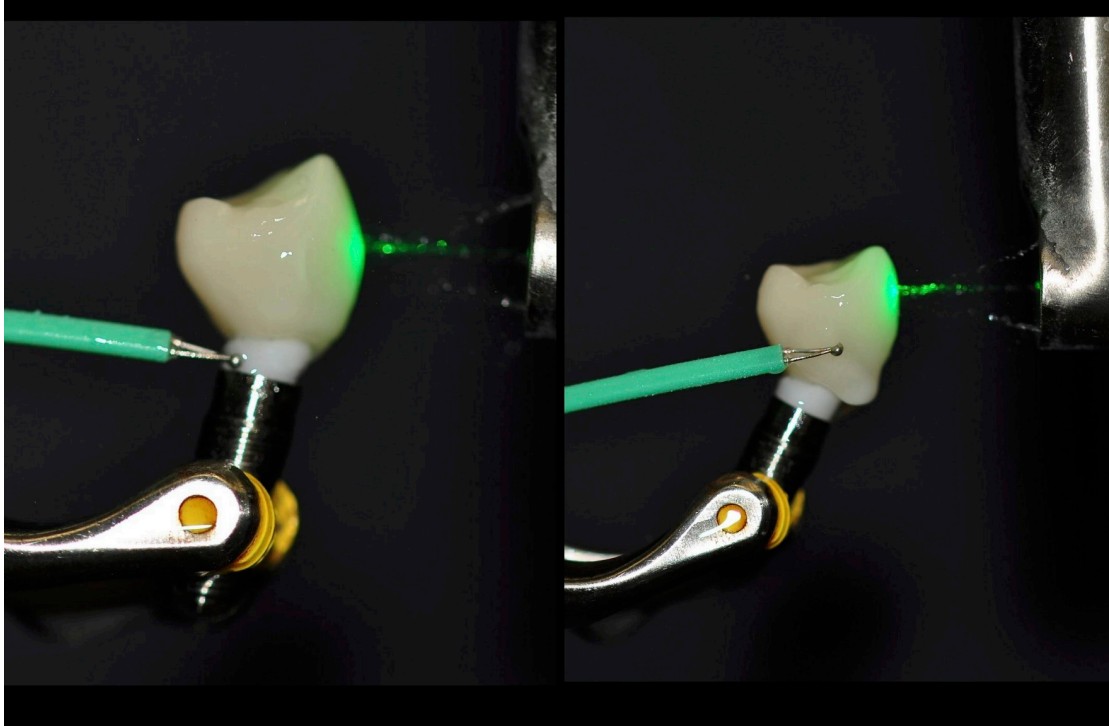

**Fig 2. Temperature measurements of crown and abutment during laser irradiation.**

**Table 1. Crown removal time (sec) and statistical analyses.**

| Grouping$ | LT | LTR | LTRR | ST |
|---|---|---|---|---|
| Mean | 282 | 204 | 192 | 192 |
| SD | 56.92 | 23.66 | 20.98 | 20.98 |
| ANOVA* | p < .001 | | | |
| | | | | |
| | t-Test*,# | | | |
| | LT-LTR | LT-LTRR | LT-ST | |
| | p < .001 | p < .001 | p < .001 | |
| | LTR-LTRR | LTR-ST | LTRR-ST | |
| | p = .246 | p = .246 | p = 1 | |

$LT: Long-term cementation, LTR: Long-term recementation, LTRR, Long-term repeated recementation, and ST: Short-term cementation

*Level of statistical significance, p<0.05

#two-sample assuming equal variances and P(T< = t) two-tail

the removal times of LTR, LTRR, and ST (t-test, p = .246, .246 and 1) or short-term cementation as well as the two recementation group (p<0.05). After irradiation, all crowns were retrieved of the abutment manually using digital manipulation. By visual inspection, no crowns/abutments appeared fractured or damaged of under 40X magnification (Leica M320, Leica Microsystems, Germany). Examination using SEM (Figs 3 and 4) showed no structural changes or damage suggestive of photoablation or thermal ablation. The cement appeared to stay mostly in the crown. Additionally, no carbonization on the zirconia implant abutment was observed. SEM analysis conducted to test the damage of the implant crown (Fig 3) and abutment surface (Fig 4) showed that all laser debonded samples demonstrated no cracks or fractures with macro and microstructure. Slight occasional ablation of the cement inside the crown during irradiation was observed only in one case in the study. The temperatures of the abutment and the crown after 1 to 10 min of irradiation are shown in Table 2. The average temperatures were 19°C for the abutment and 23.3°C for the crown. There was a statistical significant difference between the temperature of the abutment and the crown (Paired t-test, p < .0001).

## Discussion

Clinically, removing a cemented lithium disilicate crown from an implant abutment can sometimes take over 30 to 60 minutes of dental chair time. More importantly, the conventional rotary instrument crown removal technique most of the time would render the lithium disilicate crown and often underlined zirconia abutment unreusable. Lithium disilicate and zirconia materials would often be chattered or cracked upon touching by a rotary instrument. Conventionally, when clinicians need to retrieve a cement-retained single implant restoration, it almost always resulting in remaking of the crown and/or the abutment. Therefore, there is a clear need for a more conservation approach of implant crown removal. The results here suggested that Er:YAG laser can be used as a non-invasive tool to retrieve a lithium disilicate crown of a zirconia prefabricated abutment. The results further showed a long-term cemented crown, of 6 months, may be slightly more difficult to remove with Er:YAG laser than a crown with repeated cementation or new cementation. This may be a result of older cement possibly containing less residual monomer. It is also possible that the way we kept the samples allowed samples to get drier in long term storage. Missing water molecules in drier situation may result in less water activation from Er:YAG laser and thus it is more difficult to remove the crown. It

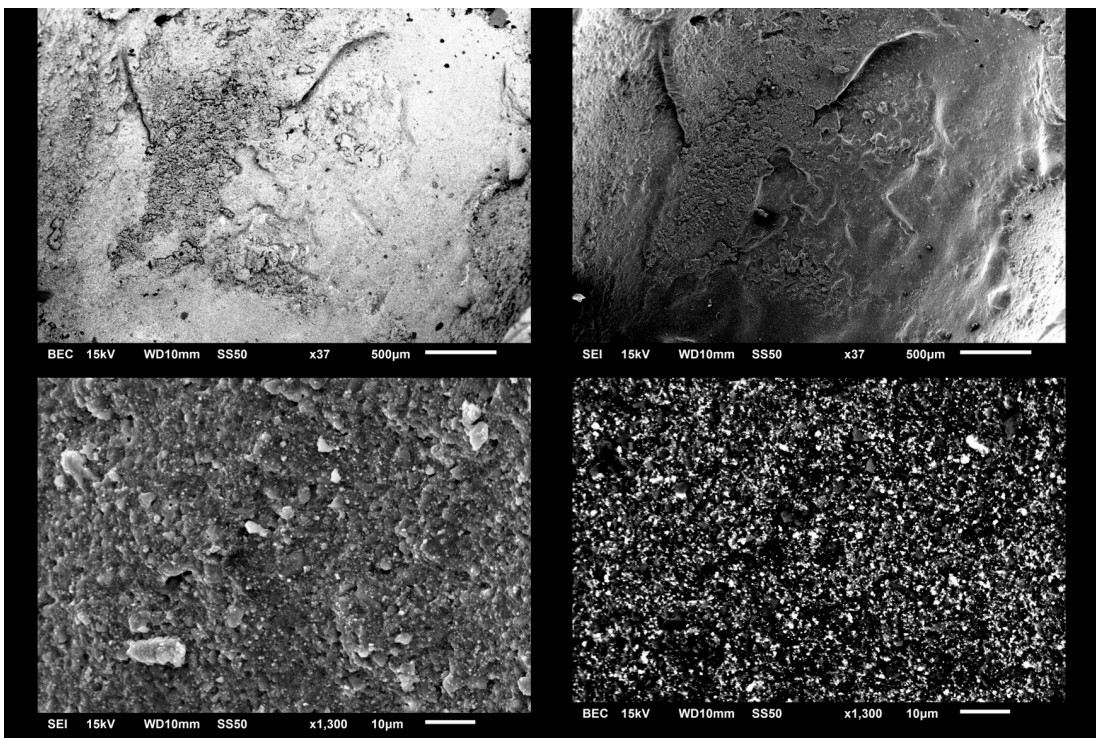

**Fig 3. SEM Images of lithium disilicate crown after irradiation at 500μm and 10μm scale, note intact surface with residual cement.** Top left and right demonstrates undamage smooth lithium disilicate material with some residual cement 500μm scale. Bottom left and right demonstrates normal crystal structure of lithium disilicate at 10μm.

is also interesting to point out that there were no statistical differences among repeated cementation and new cementation samples. This suggested that Er:YAG laser implant crown retrieval is efficient in early cementation whether or not the crown is newly or repeatedly cemented. Clinically, it would be useful to enable a clinician to retrieve the crown from an implant abutment anytime an indication presents itself.

Overall, the retrieval time range ~3–6 min which are slightly higher than ~3–4 minutes reported for a removal from a natural tooth abutment[20] but similar to the previous report of retrieval time of a lithium disilicate crown from a titanium abutment ~1:30 to 5 minutes. [22] This is possibly due to higher water content in the natural tooth abutment compared to the zirconia abutment. Also, the cementation of lithium disilicate to zirconia may have slightly stronger retention than lithium disilicate to titanium. Thus, the Er-YAG laser water molecule activation during crown removal is more efficient for a natural tooth abutment compared to zirconia implant abutment. It is important to note the laser setting differences in this study compared to Rechmann et al [20]. Rechmann et al[20] applied a higher energy setting of of 500 mJ, 10 Hz, and 5 W, for ~4–5 min compared to the energy setting in this study of 300mJ; 15Hz, and 4.5W for ~4–6 min. Note also that the crown thickness in the present study of ~ 1–2 mm (based on the computer-aided design/computer-aided manufacturing or CAD/CAM software used) was comparable to the thickness of 1.0 mm in Rechmann et al [20]. The laser setting in this study was also slightly lower than other studies. For instance, a setting of 320mJ, 20Hz under water irrigation using chisel-tip (1.2x0.4mm) was used to remove porcelain veneers[23] and a setting of 135mJ, 15Hz, 2W was used for monolithic zirconia and 200mJ, 15Hz, 3W for lithium disilicate crown from the teeth.[24] The results therefore indicate that

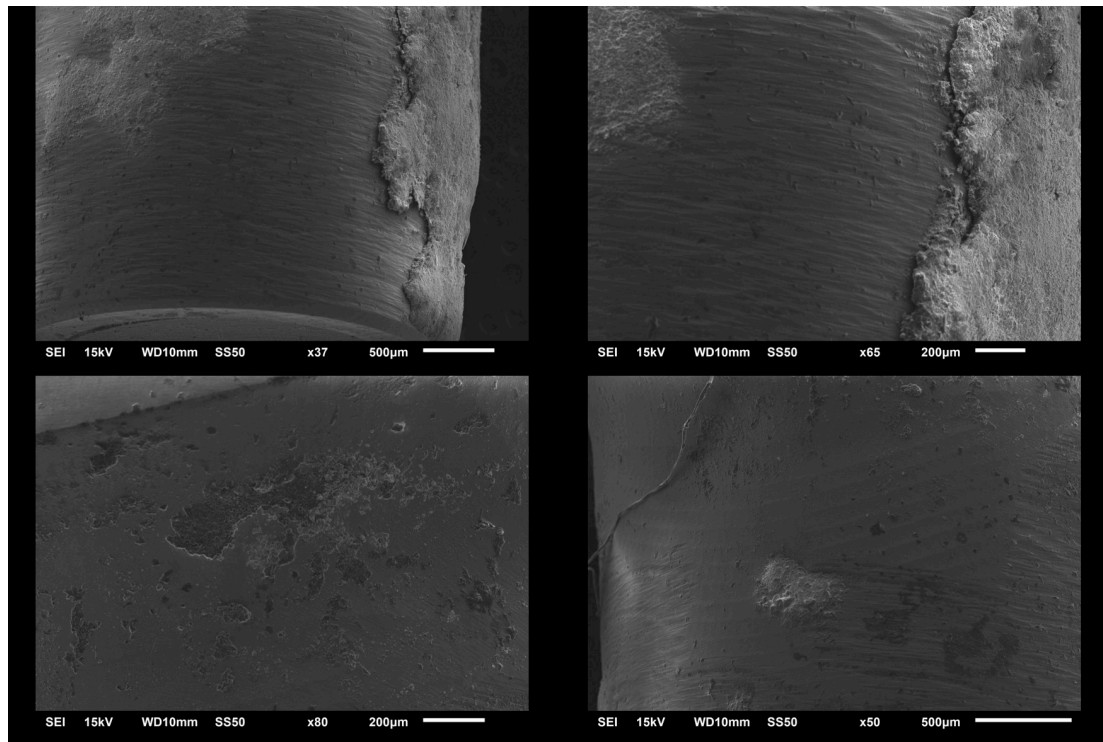

**Fig 4. SEM images zirconium abutment without irradiation and after irradiation at 200μm and 500μm scale.** Note residual cement on intact abutment surface. Top left (500μm) and right (200μm) demonstrates area of zirconia abutment surface with residual cement. Bottom left (200μm) and right (500μm) demonstrate the area of zirconia abutment without residual cement. Note normal smooth surface without any noticeable damage of zirconia.

lower energy setting in this present study may present a safe and sufficient method for implant crown removal.

**Table 2. Temperature measurements during irradiation for decementation with Er:YAG laser of the all-ceramic crowns from Zirconia abutments cemented with resin cement[#].**

| Time in minutes | Temperature (˚C) of abutment surface | Temperature (˚C) of crown surface |
|---|---|---|
| 1 | 20 | 22.2 |
| 2 | 19.2 | 23.1 |
| 3 | 19.8 | 24.5 |
| 4 | 19.2 | 23.4 |
| 5 | 18.8 | 22.9 |
| 6 | 18.4 | 23.7 |
| 7 | 19.9 | 22.7 |
| 8 | 18.4 | 23.9 |
| 9 | 18.9 | 23.0 |
| 10 | 19.1 | 23.6 |
| Average | 19.17 | 23.3 |
| Standard Deviation | 0.58 | 0.66 |
| Pair t-test[*] | p < .0001 | |

[*]P(T< = t) one-tail

[#]Ambient temperature was set at 20˚C.

Two important issues to be considered when laser is applied to retrieve a crown from an implant abutment include the damage to the crown and the possible increase in temperature to the abutment. First, there was no visual damage to any crown or abutment in the present study. No macro- or micro- structural surface damage was seen in both microscope and SEM. A study using Er,Cr:YSGG laser to remove lithium disilicate crowns from natural teeth, showed a lesser time of removal (~1–3 min) however gross damages to the crowns and natural teeth were reported.[25] This may have been a result of a different type of laser and a higher energy setting. Furthermore, the minimal damage to the retrieved lithium disilicate crown by Er:YAG laser can be explained by the radiation energy absorption pattern. The Fourier Transform Infrared Spectroscopy (FTIR), method used to measure material energy absorption, demonstrates a broad $H_2O$/OH absorption band of wavelengths in the range of 3,750–3,640 and 3,600–3,400 nm. These ranges coincide with the Er:YAG laser emission wavelength. Composite resin cements, such as Multilink (Ivoclar Vivadent) demonstrate a distinct absorption peak at ~3,401 nm. There was little radiation absorption to lithium disilicate material.[20,26] Thus, this allows debonding via irradiation energization of cement with little or no damage to the crown material.

Second, it is important to minimize the temperature increase during laser irradiation in order to prevent damage to the implant-bone interface and osseointegration. The temperatures recorded at the most apical portion of the abutment during 1–10 min of irradiation were raised less than 4 to 7˚C. The temperature increases from ~22˚C to ~25˚C for the crown are similar to previous study on a natural tooth abutment.[27] Note that the abutment temperature about ~4˚C lower. Thus, clinical damages to the abutment or implant fixture or osseointegration are unlikely. It is generally believed that the temperature >47˚C can cause irreversible damage to dental implants.[28–30] The increase in temperature in the present study was similar to the Er-YAG laser removal of a lithium disilicate crown from a natural tooth abutment. [27] and this temperature increase is considered clinically safe. Note that the maximal time of abutment crown retrieval from a titanium implant from our previous study [22] here were 5 minutes. Therefore, we did 10 minutes to double the possible maximal time of irradiation and to keep the results comparable to our previous study [22]. We do not think that the clinician will ever need to do more then 5–10 minutes irradiation at a time (often it will only be 30-second to 3-minute interval).

While this study provides a proof-of-principle and feasibility of all-ceramic-implant crown removal using Er:YAG laser, this is an *in vitro* study and some limitations should be addressed. First, there may be difficulty in accessing the implant abutment crown margin in patient's mouth. In this study, the irradiation was concentrated in the cervical area of the implant using tipless handpiece that may facilitate better clinical access compared to a handpiece with a tip. It is possible that in some clinical situation the implant may be placed deep under soft tissue and access to the crown margin may be challenging. In those cases, it could take longer to retrieve the crown. Second, the thickness of the crown can vary and be more than 1–1.5 mm used in this study. Again, more clinical time may be needed to remove the crown. Third, the cement used in this study, composite resin cement-Variolink Esthetic (Ivoclar Vivadent), may not be applicable to other cements such as resin modified glass ionomer or other cements that may have different water/monomer content and therefore different laser energy absorption. Fourth, laser operator's skills, in this study one operator (Author-KGL) who was an expert in Er:YAG laser, may affect the efficiency and thus the length of irradiation. Clinicians with varied skills in Er:YAG laser may require more or less time in crown retrieval. Finally, it is important to state here that clinical study in humans with varieties of abutment/crown materials, cements and prosthetic designs will be needed in the future to further optimize the clinical applications of Er:YAG laser. As previous mentioned in the introduction, Er:YAG laser has a

wide range of applications including the treatment of peri-implantitis, chronic inflammation of hard of soft tissue around dental implants.[31–34] Er:YAG laser has been used to decontaminate implant surface by eliminating biofilm and allow more effective bone grafting for bone defects around dental implants.[31–34] It would be interesting to see if similar applications of this laser would alter biofilm formation on the restorative surface during crown retrieving. Future clinical studies in humans as well as studies with an artificial biofilm system may provide further insight into the application of Er:YAG laser in implant and restorative dentistry.

## Conclusions

Cement-retained implant prostheses can present a clinical challenge when retrieval of the prosthesis is indicated for the treatment of peri-implantitis, retrieval of the abutment screw, and repair/remade of the prosthesis. Atraumatic decementation of cement-retained implant prosthesis using Er:YAG laser is a viable method in retrieving in short-, long-term and repetitive cemented lithium disilicate implant crowns from zirconia prefabricated abutments. Er:YAG lasers can retrieve cemented prosthesis atraumatically without damaging the abutment and prostheses. The present study provides safe parameters and predictable protocol, which could be easily applied to the routine dental practice.

## Supporting information

**S1 File. The detailed data information and statistical analyses are presented here.**
(XLSX)

## Acknowledgments

The authors thank Dr. Janina Lewis as well as the staff and students in her laboratory for experimental assistance. Thanks also to staff and faculty of the Virginia Commonwealth Univeristy Center of Digital Dentistry especially Robert Armstead, April Harris, Marithe Blacagon, and JoAnn Marreo for their support in sample preparation and experimental assistance.

## Author Contributions

**Conceptualization:** Janina Golob Deeb, Sompop Bencharit, Kinga Grzech-Leśniak.

**Data curation:** Sompop Bencharit, Kinga Grzech-Leśniak.

**Formal analysis:** Sompop Bencharit, Kinga Grzech-Leśniak.

**Investigation:** Janina Golob Deeb, Sompop Bencharit, Nishchal Dalal, Aous Abdulmajeed.

**Methodology:** Sompop Bencharit, Nishchal Dalal, Aous Abdulmajeed.

**Project administration:** Janina Golob Deeb.

**Visualization:** Sompop Bencharit.

**Writing – original draft:** Sompop Bencharit.

**Writing – review & editing:** Janina Golob Deeb, Sompop Bencharit, Aous Abdulmajeed, Kinga Grzech-Leśniak.

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
