## [Decision Letter · Decision Letter 0]

27 Aug 2019

PONE-D-19-18100

Using Er:YAG Laser to remove lithium disilicate crowns from zirconia implant abutments: an in vitro study

PLOS ONE

Dear Dr. Bencharit,

Thank you for submitting your manuscript to PLOS ONE. After careful consideration, we feel that it has merit but does not fully meet PLOS ONE’s publication criteria as it currently stands. Therefore, we invite you to submit a revised version of the manuscript that carefully addresses the points raised during the review process.

We would appreciate receiving your revised manuscript by Oct 11 2019 11:59PM. To enhance the reproducibility of your results, we recommend that if applicable you deposit your laboratory protocols in protocols.io, where a protocol can be assigned its own identifier (DOI) such that it can be cited independently in the future. For instructions see: http://journals.plos.org/plosone/s/submission-guidelines#loc-laboratory-protocols

We look forward to receiving your revised manuscript.

Kind regards,

Yogendra Kumar Mishra, Ph. D.

Academic Editor

PLOS ONE

**Journal Requirements:**

2. We noticed you have some minor occurrence of overlapping text with previous publications, which needs to be addressed. In your revision ensure you cite all your sources (including your own works), and quote or rephrase any duplicated text outside the methods section.

**Comments to the Author**

1. Is the manuscript technically sound, and do the data support the conclusions?

Reviewer #1: Yes

Reviewer #2: Yes

2. Has the statistical analysis been performed appropriately and rigorously? 

Reviewer #1: Yes

Reviewer #2: Yes

3. Have the authors made all data underlying the findings in their manuscript fully available?

Reviewer #1: Yes

Reviewer #2: Yes

4. Is the manuscript presented in an intelligible fashion and written in standard English?

Reviewer #1: Yes

Reviewer #2: Yes

5. Review Comments to the Author

Reviewer #1: In manuscript PONE-D-19-18100, Bencharit et al. evaluated Er:YAG laser to remove lithium disilicate crowns from zirconia implant abutments. The work is continuation of authors’ previous work on retrieval of lithium disilicate crowns from titanium implant abutments. The experiments are interesting and the results are well interpreted with statistical analysis. The manuscript is recommended for publication in PLOS One, however authors are suggested to include the p-values in table 1 and 2 instead of just reporting it as p< 0.001 for ANOVA test.

Reviewer #2: The manuscript explains very well evaluate the effect of erbium-doped yttrium aluminum garnet (Er:YAG) laser as a non-invasive treatment modality to remove lithium disilicate crowns from zirconia implant abutments. This publication can be published, and requires minor revision with the focus on comments below:

1. The study is interesting, were there any short-term recementation and repeated recementation studies performed too?

2. The abbreviations need to be expanded when they appear in the text after the first time.

3. What was the rationale behind studying the temperature changes of the abutment and crown only upto 10 minutes? Views on beyond 10 min time point?

4. Please elaborate a little more on how Er:YAG laser usage, allows more reusage of lithium disilicate crowns.

5. Introduction First Paragraph, last line: Zirconia abutments offer white esthetics……Please explain white esthetics.

6. “Er:YAG lasers can retrieve cemented prosthesis a traumatically without damaging the abutment and prostheses”. Are there any future perspectives of employing Er:YAG lasers in other teeth and gums related diseases

6. Figures 3 and 4, please elaborate on the legend, and explain each quadrant as A) B) C) D) and their significance.

7. Grammatical errors and sentence errors should be corrected throughout the manuscript.

8. Authors insights or future perspective on in-vivo studies using Er:YAG lasers

Overall it is a well written manuscript. Explaining clearly the importance of Er:YAG lasers. If Published, it will be of valuable importance.

6. PLOS authors have the option to publish the peer review history of their article (what does this mean?). If published, this will include your full peer review and any attached files.

Reviewer #1: No

Reviewer #2: No

---

## [Author Response · Author response to Decision Letter 0]

28 Aug 2019

Journal Requirements:

RESPONSE: The manuscript was amended per PLOS ONE’s style requirements.

2. We noticed you have some minor occurrence of overlapping text with previous publications, which needs to be addressed. In your revision ensure you cite all your sources (including your own works), and quote or rephrase any duplicated text outside the methods section.

RESPONSE: The work is a follow up study from our previous work, Reference #22. The texts outside the methods were checked for potential plagiarism. Appropriate citations were made.

Reviewer #1: In manuscript PONE-D-19-18100, Bencharit et al. evaluated Er:YAG laser to remove lithium disilicate crowns from zirconia implant abutments. The work is continuation of authors’ previous work on retrieval of lithium disilicate crowns from titanium implant abutments. The experiments are interesting and the results are well interpreted with statistical analysis. The manuscript is recommended for publication in PLOS One, however authors are suggested to include the p-values in table 1 and 2 instead of just reporting it as p< 0.001 for ANOVA test.

RESPONSE: The authors truly appreciate this kind insight and comments from the reviewer. Please kindly find our explanation below.

The calculated ANOVA value for Table 1 is 0.0000008534387269

and the calculated t-test values of LT-LTR, LT-LTRR, and LT-ST were 0.0008373721939, 0.0001817757697, 0.0001817757697, respective. However, we can only report as the p values as low as p < 0.001 per the conventional statistical report decimal point for the ANOVA, LT-LTR, LT-LTRR, and LT-ST. 

Reviewer #2: The manuscript explains very well evaluate the effect of erbium-doped yttrium aluminum garnet (Er:YAG) laser as a non-invasive treatment modality to remove lithium disilicate crowns from zirconia implant abutments. This publication can be published, and requires minor revision with the focus on comments below:

1. The study is interesting, were there any short-term recementation and repeated recementation studies performed too?

RESPONSE: The authors appreciate this comment. However, in our experimental design, we attempted to mimic clinical situations when clinicians often do not attempt to repeated retrieval/cementation at the short-term cementation (ST). And if so, we assume that the retrieval time of the short-term repeated would be similar to the repeated long-term cementation (LTRR) since the ST and LTRR groups have no statistical difference between them (p=1).

2. The abbreviations need to be expanded when they appear in the text after the first time.

RESPONSE: The authors appreciated this comment. Abbreviations, such as Er:YAG, Er,Cr:YSGG, ANOVA, CAD/CAM were explained at the first time mentioned in the maintext.

3. What was the rationale behind studying the temperature changes of the abutment and crown only upto 10 minutes? Views on beyond 10 min time point?

RESPONSE: We appreciated this comment. The following rationale was added into the Discusson.

“Note that the maximal time of abutment crown retrieval from a titanium implant from our previous study [22] here were 5 minutes. Therefore, we did 10 minutes to double the possible maximal time of irradiation and to keep the results comparable to our previous study [22]. We do not think that the clinician will ever need to do more then 5-10 minutes irradiation at a time (often it will only be 30-second to 3-minute interval).”

4. Please elaborate a little more on how Er:YAG laser usage, allows more reusage of lithium disilicate crowns.

RESPONSE: This Is a truly great insight. The following statement was added at the beginning of the Discussion.

“Clinically, removing a cemented lithium disilicate crown from an implant abutment can sometimes take over 30 to 60 minutes of dental chair time. More importantly, the conventional rotary instrument crown removal technique most of the time would render the lithium disilicate crown and often underlined zirconia abutment unreusable. Lithium disilicate and zirconia materials would often be chattered or cracked upon touching by a rotary instrument. Conventionally, when clinicians need to retrieve a cement-retained single implant restoration, it almost always resulting in remaking of the crown and/or the abutment. Therefore, there is a clear need for a more conservation approach of implant crown removal.”

5. Introduction First Paragraph, last line: Zirconia abutments offer white esthetics……Please explain white esthetics.

RESPONSE: The clarification statement was added. The statements are now read as followed.

“Conventionally, titanium abutments are often used as an intermediate between the dental implants. Like other metals, titanium abutments can exhibit greyness or dark shade through a translucent lithium disilicate crown. Zirconia abutments are therefore often prescribed to offer white esthetics. Zirconia abutments have therefore an advantage over the metal abutment especially in patients with thin gingival biotype.[2,3],[8],[9]”

6. “Er:YAG lasers can retrieve cemented prosthesis a traumatically without damaging the abutment and prostheses”. Are there any future perspectives of employing Er:YAG lasers in other teeth and gums related diseases.

RESPONSE: The authors appreciated this comment. The following statement was added.

“As previous mentioned in the introduction, Er:YAG laser has a wide range of applications including the treatment of peri-implantitis, chronic inflammation of hard of soft tissue around dental implants.[31-34] Er:YAG laser has been used to decontaminate implant surface by eliminating biofilm and allow more effective bone grafting for bone defects around dental implants.[31-34] It would be interesting to see if similar applications of this laser would alter biofilm formation on the restorative surface during crown retrieving. Future clinical studies in humans as well as studies with an artificial biofilm system may provide further insight into the application of Er:YAG laser in implant and restorative dentistry.”

6. Figures 3 and 4, please elaborate on the legend, and explain each quadrant as A) B) C) D) and their significance.

RESPONSE: The figure legends were amended (top/bottom and left/right were used instead of A-D).

Figure 3: SEM Images of lithium disilicate crown after irradiation at 500µm and 10µm scale, note intact surface with residual cement. Top left and right demonstrates undamage smooth lithium disilicate material with some residual cement 500µm scale. Bottom left and right demonstrates normal crystal structure of lithium disilicate at 10µm.

Figure 4: SEM images zirconium abutment without irradiation and after irradiation at 200µm and 500µm scale. Note residual cement on intact abutment surface. Top left (500µm) and right (200µm) demonstrate area of zirconia abutment surface with residual cement. Bottom left (200µm) and right (500µm) demonstrates the area of zirconia abutment without residual cement. Note normal smooth surface without any noticeable damage of zirconia.

7. Grammatical errors and sentence errors should be corrected throughout the manuscript.

RESPONSE: Corrections have been made.

8. Authors insights or future perspective on in-vivo studies using Er:YAG lasers

RESPONSE: The following statement was added.

“It would be interesting to see if similar applications of this laser would alter biofilm formation on the restorative surface during crown retrieving. Future clinical studies in humans as well as studies with an artificial biofilm system may provide further insight into the application of Er:YAG laser in implant and restorative dentistry.”

Overall it is a well written manuscript. Explaining clearly the importance of Er:YAG lasers. If Published, it will be of valuable importance.

 RESPONSE: We appreciated this kind comment.

---

## [Decision Letter · Decision Letter 1]

2 Oct 2019

Using Er:YAG Laser to remove lithium disilicate crowns from zirconia implant abutments: an in vitro study

PONE-D-19-18100R1

Dear Dr. Bencharit,

We are pleased to inform you that your manuscript has been judged scientifically suitable for publication and will be formally accepted for publication once it complies with all outstanding technical requirements.

With kind regards,

Yogendra Kumar Mishra, Ph. D.

Academic Editor

PLOS ONE

Additional Editor Comments (optional):

Reviewers' comments:

Reviewer's Responses to Questions

**Comments to the Author**

1. If the authors have adequately addressed your comments raised in a previous round of review and you feel that this manuscript is now acceptable for publication, you may indicate that here to bypass the “Comments to the Author” section, enter your conflict of interest statement in the “Confidential to Editor” section, and submit your "Accept" recommendation.

Reviewer #1: All comments have been addressed

Reviewer #2: All comments have been addressed

2. Is the manuscript technically sound, and do the data support the conclusions?

Reviewer #1: Yes

Reviewer #2: Yes

3. Has the statistical analysis been performed appropriately and rigorously? 

Reviewer #1: Yes

Reviewer #2: Yes

4. Have the authors made all data underlying the findings in their manuscript fully available?

Reviewer #1: Yes

Reviewer #2: Yes

5. Is the manuscript presented in an intelligible fashion and written in standard English?

Reviewer #1: Yes

Reviewer #2: Yes

6. Review Comments to the Author

Reviewer #1: The authors have addressed all the questions and manuscript is recommended to be accepted for publication.

Reviewer #2: The authors have addressed the questions and comments, and have explained the queries very well. They have additionally made necessary changes and have met the requirements pointed in the comments section.

7. PLOS authors have the option to publish the peer review history of their article (what does this mean?). If published, this will include your full peer review and any attached files.

Reviewer #1:

Reviewer #2:

---

## [Editor Report · Acceptance letter]

25 Oct 2019

PONE-D-19-18100R1 

Using Er:YAG Laser to remove lithium disilicate crowns from zirconia implant abutments: an *in vitro* study 

Dear Dr. Bencharit:

I am pleased to inform you that your manuscript has been deemed suitable for publication in PLOS ONE. Congratulations! Your manuscript is now with our production department. 

With kind regards,

on behalf of

Dr. Yogendra Kumar Mishra 

Academic Editor

PLOS ONE